# Incorporating Consumer Insights into the UK Food Packaging Supply Chain in the Transition to a Circular Economy

**Nikki Clark \*** **, Rhoda Trimingham and Garrath T. Wilson**

School of Design and Creative Arts, Loughborough University, Loughborough LE11 3TU, UK;
R.L.Trimingham@lboro.ac.uk (R.T.); G.T.Wilson@lboro.ac.uk (G.T.W.)
\* Correspondence: n.clark@lboro.ac.uk

**Abstract:** The growth of eating lunch purchased out of the home has led to an increased need for pre-packaged food-to-go products. Single-use plastic packaging is frequently chosen for its food safety and convenience attributes; however, the material format is under scrutiny due to concerns over economic waste and environmental impact. A circular economy could transform linear make-use-dispose supply chains into circular systems, ensuring the cycling of valuable plastic resources. However, there has been limited research into how consumers will behave within circular economic systems. Understanding consumer behaviour with packaging disposed out of the home could aid designers in developing solutions society will adopt in the transition to a circular economy. This study evaluates the application of behaviour research methods, and the behavioural insight outputs, with stakeholders from the UK food-to-go packaging supply chain. A novel co-design workshop and business origami technique allowed multiple stakeholder groups to collaboratively discuss, evaluate, and plan how consumer behaviour techniques could be used within their supply chain packaging development process. Although all stakeholders identified strengths in incorporating behaviour studies into the development process, providing essential knowledge feedback loops, barriers to their application include the cost and time to implement, plus the existing inconsistent UK waste infrastructure.

**Keywords:** consumer behaviour; circular economy; food packaging; sustainable design; circular supply chains

---

## 1. Introduction

An increase in the UK's convenience lifestyle has led to significant growth of the retail food-to-go (FTG) market, valued at £2.9 billion in 2019 [1], with a further £2 billion growth forecasted by 2022 [2]. With three in five consumers in the UK regularly eating lunch out of the home [3], 16–44-year olds are most likely to have lunch on the move [4]. The popularity for UK workers to buy lunch out of the home has grown in the last five years, self-attributed predominately to a time poor lifestyle [5]. Research conducted by environmental charity Hubbub Foundation UK found, on average, workers purchased four packaged items for lunch, generating an estimated 10.7 billion separate items of waste per year [5]. Single-use plastic packaging forms a significant proportion of this waste, and since 2016 concerns have been raised about the global economic and environmental impact of plastic packaging waste [6]. Pre-dating the BBC's 2017 Blue Planet series, which in turn heightened public awareness leading to the European Commission's 'Single-Use Plastics Directive', reports showed that after a short first use cycle, 95% of plastic packaging material value is lost [6]. Post-Blue Planet, the social and environmental case has been amplified, with plastic pollution cited by 47% of UK adults as the "most important environmental issue" [7].

The action-awareness gap that arises between the increase in single-use plastic packaging and the increase in consumer pro-environmental awareness is a significant challenge for the UK's packaging supply chain. However, plastic packaging also plays an essential role in the safe delivery of FTG products through the supply chain to the consumer, offering a convenient way to eat on the go. The packaging's structural design features allow ease of access, product visibility, handling, and disposal [8], and all have a part to play in making the pack convenient for the consumer to use, durable, affordable, and enhancing the eating experience [9]. From design conception to end-of-life disposal, food packaging goes through a complex lifecycle, alongside the food product's lifecycle, involving multiple stakeholders [10]. Important decisions must be made at each point in the packaging lifecycle, to ensure that the pack safely and efficiently performs, minimizing the waste of both packaging and food materials. It is not feasible, therefore, to simply remove all plastic products from the existing supply and demand system without significant change, but there are ways forward.

The Department for the Environment, Food, and Rural Affairs (DEFRA) outlines a four-point strategy in its 25 Year Environment Plan, which aims to act at every stage of the product lifecycle – production, consumption, and end of life [11]. One key area is to increase resource efficiency and reduce pollution and waste, leading to a society where resources are fully valued. The UK Retail Industry faces the possible introduction of an extended producer responsibility (EPR) tax and further legislative changes that could impact on the commercial viability of using single-use plastic packaging. Outside of policy-only intervention, the circular economy (CE) has been hailed as a possible citizen-centric solution to establish the cyclical flow of plastics ensuring their value is retained for as long as possible. A scoping study established a willingness from the UK packaging industry to engage with CE concepts, however, stakeholders highlighted the commercial viability of systems, speed to implement technologies, and the unintended environmental consequences as major risks [10]. Nevertheless, there is progress; for the first time, this study showed there is industry agreement that the food packaging supply chain needs to work collaboratively and not as independent concerns. However, no major studies exist that have considered the necessary supply chain values and consumer behaviours that would specifically enable the transition of FTG packaging into a CE.

Design for CE has aimed to develop circular business models for the food packaging industry to slow resource loops and include more feedback loops within the development process. Niedderer et al. [12] discussed how within the design process it is crucial to understand user behaviour in context, especially when this behaviour leads to undesirable actions, such as throwing recyclable waste into general waste bins. A transforming role of design within CE is evident, with a change in designer's skillset required to accommodate this systems approach of thinking. Therefore, understanding the consumers' needs is pivotal to developing future products, packaging, services, and positive user experiences. Retailers and suppliers must build "stronger emotional connections" with consumers [13] (p.12) to build trust and change consumptive behaviour. Although briefly considered by De Koeijer et al. and Niero et al., another issue rarely discussed in literature is the impact moving towards CE approaches will have on the UK consumer and what they will be willing to adopt [14–17]. Recently, the authors carried out research that focused on FTG packaging used out of the home and found that a significant percentage of consumers were placing recyclable packaging items into incorrect recycling bins or general waste [18]. These examples act to further highlight the consumer as a key stakeholder in the packaging lifecycle. However, creating meaningful change in the way consumers engage with FTG packaging needs critical understanding of the cognitive and contextual factors that underpin and shape disposal behaviour.

Existing complex social and psychological structures, such as Triandis' TIB, Michie's COM-B System, and Wilson's Augmented Model of Behaviour, map the behavioural actions of consumers with the aim to determine interventions to adapt consumer behaviour, leading to more sustainable actions [19–21]. To understand the user (in the context of behaviour change), existing research methods apply a range of techniques to gain insight into user's attitude (what they say and think), and their behaviours (what they do) [22]. These research techniques, often applied within the health sector,

include surveys, interviews, ethnographic studies, and the use of empathic tools [22,23]. A technique developed by the lead author prior to this study, applied a remote ethnographic study using digital tools allowing consumers to self-document their actions and thoughts in relation to their FTG packaging interactions over a period of several weeks [24]. The data generated from this intensive study was further investigated through semi-structured interviews and an adapted Self Report Habit Index questionnaire [25], forming a mixed-methods approach to understanding consumer behaviour with packaging disposed out of the home. The outcomes of this prior research study are illustrated in Figure 1, with the three main themes identified as: Disposal knowledge, waste disposal behaviour and awareness, and packaging and its context. The subthemes within each of these three themes were explored to determine which of the three main determinants; behavioural factors, contextual factors, or interplay factors, they aligned to, as evidenced in the colour coding of Figure 1. This research yielded a rich depth of understanding framed by behavioural theory (informed by Theory of Interpersonal Behaviour and model of habits [26,27]) to establish the relationship between observed practice (through the remote digital tools) and reported action.

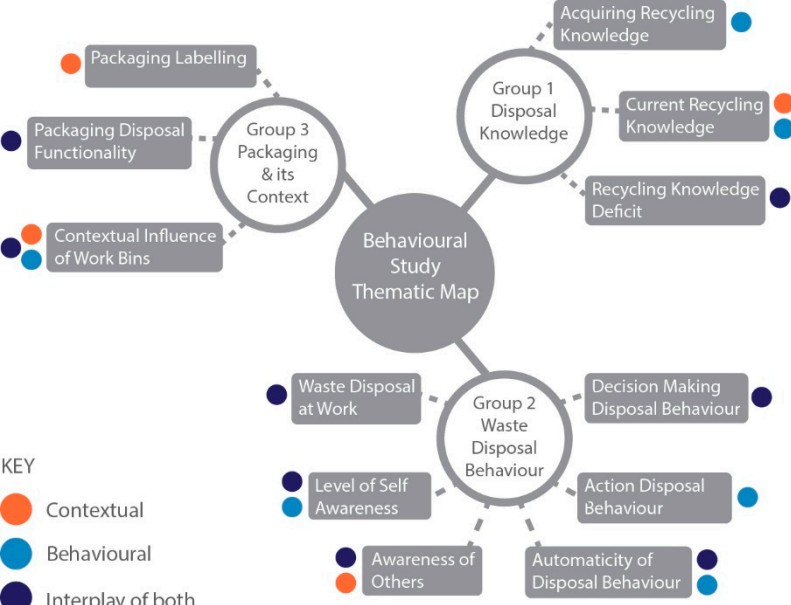

**Figure 1.** A thematic map visualising the outcomes of a consumer disposal behaviour study conducted prior to this work.

The cognitive dissonance of consumers, their perception of plastic, and growing convenience lifestyle trends can create a challenging conundrum for the food packaging supply chain. With few exceptions, NGO and governmental CE intervention strategies tend to focus on single-issue engineering aspects of the CE, such as materials manufacturing, novel recycling, and re-manufacture processes. Yet, recent research suggests it is cultural, not technological, barriers that prohibit the transition to a CE. The role of design in involving users, influencing behaviour, collaborating with industry, and improving product and service-delivery is increasingly recognised as key to successful circular solutions. Within FTG packaging development for a CE, the role of design to assist in the change of behaviour has yet to be defined. There is an opportunity to explore whether arming packaging development teams with consumer behaviour knowledge can lead to innovative solutions, providing behavioural intervention for positive behavioural change within the UK FTG context.

Further knowledge is required about the role of the user within CE design, leading to consumption changes and their impact on societies, and how this thinking can be included within the packaging development process. The aim of this research study was to understand how packaging development stakeholders can apply consumer behaviour research methods within the packaging development

process to aid the UK's FTG supply chain in the transition to a circular economy. This research will help establish consumer behaviour and value-chain focused, design-led multi-disciplinary research as a key approach to tacking plastic waste, building an area of niche expertise in the UK through the collaboration of industry stakeholders and academic experts.

## 2. Methods

A co-design workshop with 5 to 20 industry stakeholders with a good knowledge of the business ecosystem across the FTG packaging supply chain was used to evaluate the application of consumer behaviour insights within the development process [28]. Participants were selected from across five industry stakeholder groups due to their key involvement and decision-making influence within FTG product development. Stakeholders all worked within a leading UK FTG retailer's supply chain, supplying the Retail stakeholder with food products, FTG packaging of various formats, or waste management service advice. Using the lead author's prior knowledge of professionals within the UK food packaging supply chain invitation emails were sent to 23 industry professionals with 15 agreeing to attend the workshop. The participants represented packaging development roles across the Retail stakeholder's FTG supply chain (Table 1), giving a broad representation of industry stakeholders within the workshop.

**Table 1.** Job roles and supply chain group of stakeholder participants.

| Participant Code | Job Role | Stakeholder Group | Workshop Group |
|---|---|---|---|
| RET01 | Senior Packaging Technologist | Retailer company A | Both |
| FPR01 | Product Development Controller | Food Processor company B | Red |
| FPR02 | Packaging Development Manager | Food Processor company B | Blue |
| FPR03 | Insight Manager | Food Processor company B | Blue |
| FPR04 | Packaging Product Developer (FTG) | Food Processor company B | Red |
| FPR05 | Packaging Development Manager | Food Processor company C | Blue |
| FPR06 | Development Controller | Food Processor company C | Red |
| FPR07 | Packaging Development Manager | Food Processor company D | Red |
| PMN01 | Senior Business Development Manager | Packaging Manufacturer company E | Blue |
| PMN02 | Head of Construction Design | Packaging Manufacturer company E | Red |
| PMN03 | European Conceptual Design Manager | Packaging Manufacturer company F | Blue |
| PMN04 | Product Design Manager (Food Packaging) | Packaging Manufacturer company G | Blue |
| PMN05 | Strategic Business Director | Packaging Manufacturer company G | Red |
| REC01 | Head of Recycling Assets | Waste Management company H | Blue |
| NGO01 | Projects Manager | NGO company I | Red |

The insights used for the stakeholder evaluation were gained using a remote-ethnographic mixed-methods study conducted prior to this study, as outlined in the previous section. A co-design workshop method was selected to promote a collaborative stakeholder environment, which was identified as one of the CE enablers within a Scoping Study [10]. The co-design workshop was conducted during a three-hour session, held at Loughborough University's London campus in January 2020. Due to the size of the sample, the stakeholders were split into two groups of seven participants,

with equal numbers of stakeholders from each representative supply chain group. The one retailer representative moved between the groups so they could be involved in both discussions.

The objectives for the workshop were to:

1.  Understand the current development process for FTG packaging including at what stages they engage with consumers.
2.  Evaluate how the consumer behaviour methods could be applied within the FTG development process, their strengths, and weaknesses and how they could be used.

The Co-design Workshop consisted of three stages, some with multiple tasks, aligning to the four interrelated co-design engagement questions outlined by Zamenopoulos and Alexiou [29]. The workshop stages are outlined in Table 2, which explains the task objective, the timings, and a justification of why the stage was used within the workshop.

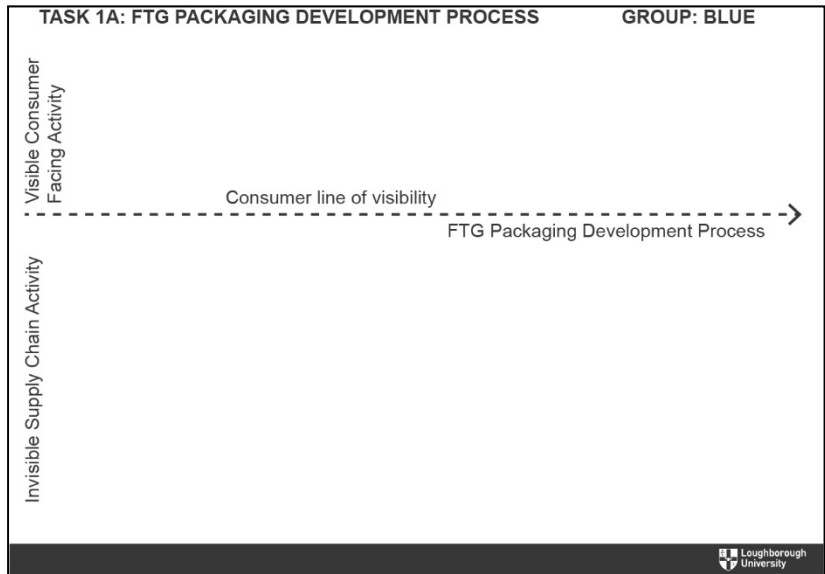

**Figure 2.** Example of a Task 1A business origami map given to each group to complete.

As part of the workshop, a desktop system mapping ('business origami') method was used [28]. This method aids the understanding of "complex value networks using simple paper cut outs representing key people, locations, channels and touchpoints" [28] (p.152). It is an interactive, rapid process allowing 5 to 15 team members to engage in the task, simply adding the cut-out icons to the map or drawing onto it. This technique is ideal for co-designing a new system but can also be used to understand how existing systems operate, aligning well with the aims of this study. The paper map was prepared prior to the workshop incorporating a process timeline and dotted line indicating whether the process stage was visible to the consumer or a behind the scenes supply chain stage, see Figure 2.

Each stakeholder group was asked to analyse their existing development process and then review it later in the workshop, indicating how they would incorporate consumer behaviour research into the process. The cut outs were placed on a large piece of paper where the different stages, interactions, and flows between system elements are drawn, see Figure 3. A colour-coded key was used to identify each stakeholder group on the process map.

**Table 2.** Stages of the co-design workshop.

| Stage/Task | Objective | Timescale | Justification of Use |
|---|---|---|---|
| **1A** | **Understanding the current process.** Each stakeholder group was asked to imagine they were developing a new piece of FTG packaging, and as a group map the existing process from design brief through to packaging launch identifying the stakeholder actions at each stage. | 40 min | The use of the provided business origami map (Figure 2) and coloured icons allowed a collaborative way for the stakeholders to discuss and plot the current FTG packaging development process. |
| **1B** | **Exploring current packaging user trials.** Stakeholders were asked to plot using a yellow post-it note where current user trials are conducted in the process. They were asked if there is a need to better understand how consumers dispose of packaging out of the home. | 15 min | The identification of existing user trials in the process allowed a comparison with later tasks on future consumer behaviour testing. Collating their YES or NO answers on a Group poster (Figure A11) provided a clear visual representation. |
| **2** | **Research Method and Findings** In this stage the Investigator described the remote-ethnography and mixed-methods approach used to gain the consumer behaviour insights and introduced the ten behavioural insight sheets, see Appendix A Figures A1–A10. | 30 min | Allowed the stakeholders to firstly understand the methods used to create the insight sheets and then introduced them to the contextual, behavioural and interplay insights they were asked to evaluate. |
| **3A** | **Analysing the Insights** Using the provided task table (Figure A12) each group were asked to evaluate five insights (Appendix A), stating reasons for it being interesting to the packaging development process, or not. | 30 min | Each group was given an equal mix of the insight sheets. This task allowed the stakeholders to discuss their interest and concern towards the insights with their thoughts captured on a prepared table to assist the analysis. |
| **3B(i)** | **Adapting the development process to explore consumer disposal behaviour** This task aimed to find out how the participants could apply the consumer behaviour insights within the FTG development process. Using their business origami map from Stage 1 to capture their ideas. | 10 min | Stakeholders were asked to add their ideas for including behavioural insight methods, establishing where and when in the development process and in what mechanism or format such ideas could be included. |
| **3B (ii)** | **Evaluating the strengths and weaknesses of incorporating behavioural insights into the FTG development process.** Stakeholders ideas were captured on post-it notes and added to each Group's task poster (Figure A13). | 5 min | Each stakeholder was asked to consider the strengths and weaknesses of incorporating behavioural insights into the FTG packaging development process. Justifying their decision. |

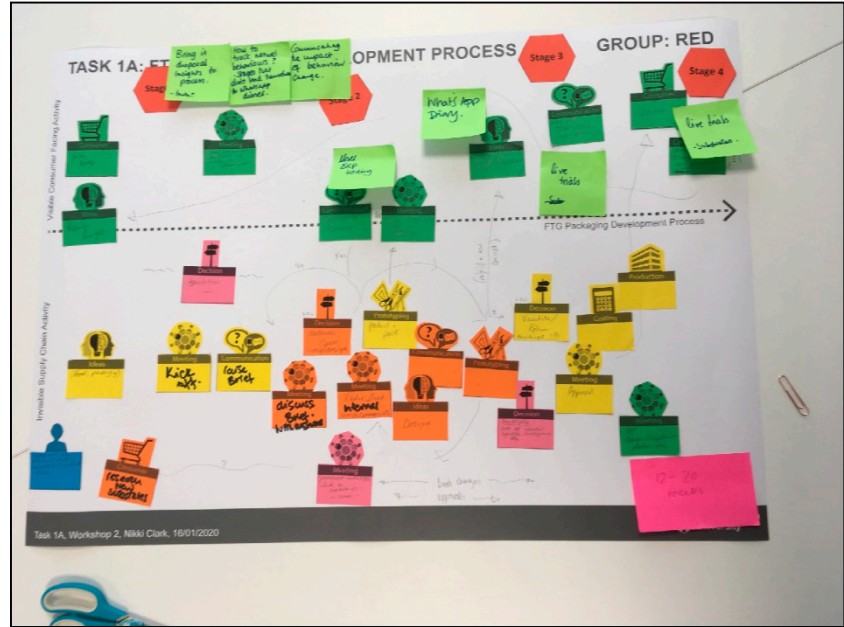

**Figure 3.** Example of a business origami map completed during the study. Icon key: Green (retailer), yellow (food processor), orange (packaging manufacturer), blue (recycler), and pink (other stakeholder).

Data were collected in the forms of business origami maps (representing current development process, pain points, where decisions are made and where user trials should be applied); audio and video recording of the discussion between packaging development practitioners were taken during the workshop; and other tangible outputs created from the workshop tasks. Thematic analysis was used to analyse this qualitative data set. Within Braun and Clarke's version of reflexive thematic analysis the "subjectivity of the researcher is seen as integral to the process of analysis," [30]. This was an important consideration within this research when the co-designing researcher brings prior knowledge of packaging development to the study. An inductive approach to coding and theme development was used, allowing a more flexible process with the coding improving as the researcher immerses themselves further within the data [30]. Affinity Diagramming is an inductive thematic analysis technique; in this approach, the themes are discovered from the data collected [31]. It helps designers move from 'data' to 'information' by visualising links, patterns, and connections between statements from different users [32]. Terry et al. describe this stage of analysis as a "productive, iterative, reflective process of data-engagement" [30].

## 3. Results and Findings

The results and findings from the three stages of the workshop that relate to the objectives set out in the previous section will be explained in this section, defining the existing consumer engagement in the FTG packaging development process, the need to better understand how consumers dispose of FTG packaging out of the home, and the stakeholder evaluation of how consumer behaviour methods could be applied.

### 3.1. Existing Consumer Engagement in the FTG Packaging Development Process

During Task 1A of the workshop, both groups plotted the current packaging development process onto a business origami map; see Figures 4 and 5, which have been redrawn for clarity.

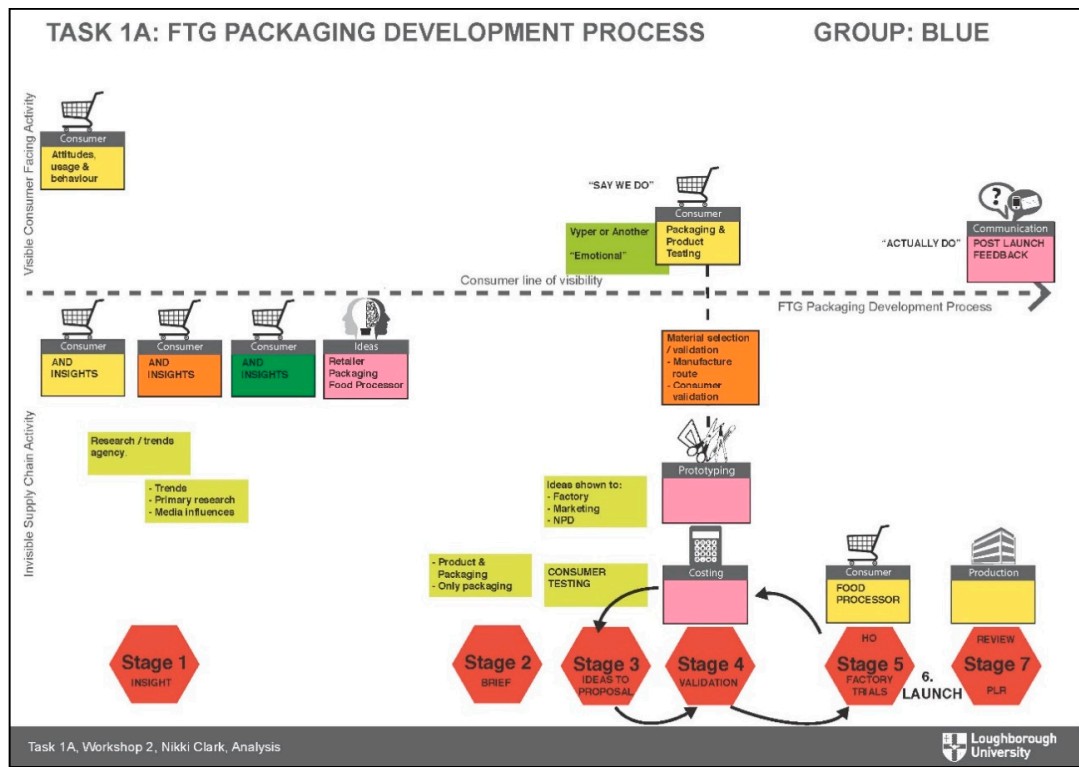

**Figure 4.** Blue group's business origami map completed in Stage 1.

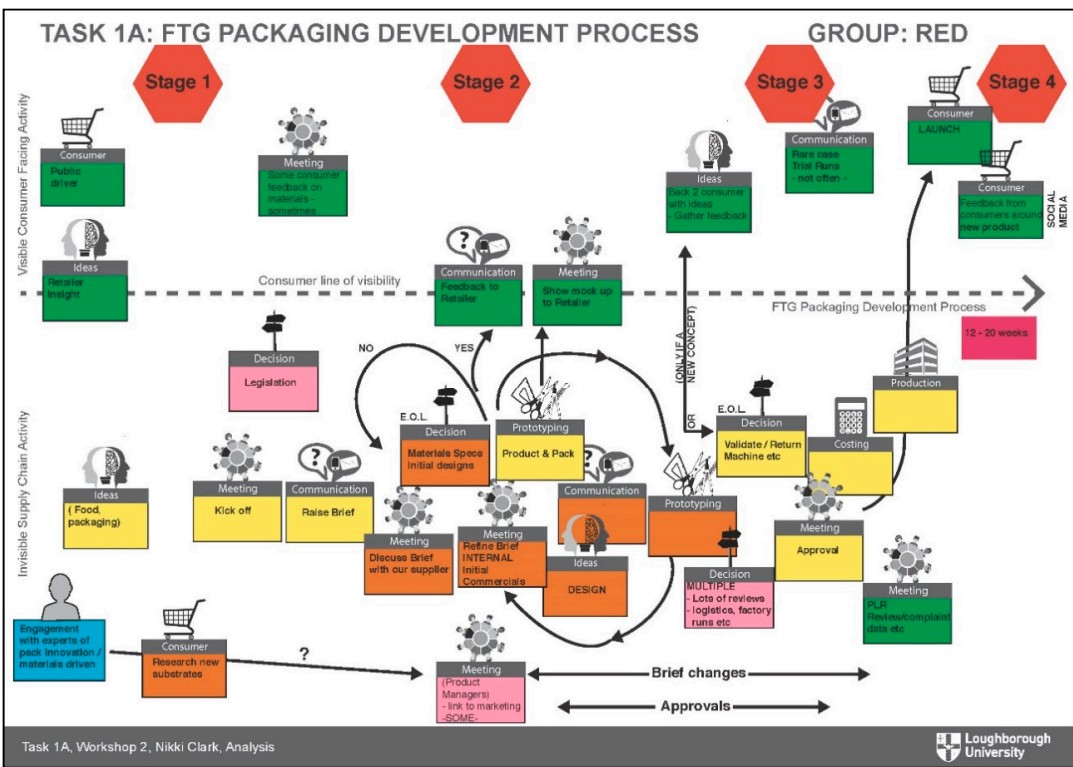

**Figure 5.** Red group's business origami map completed in Stage 1.

Although the Red Group stated there were four stages and the Blue Group stated there were seven, during the analysis stage it was found there were six individual stages discussed by the groups during the task. Stage 1 was split into two different ways to formulate a design brief (see Figure 6). Stage 1A,

1B, and 2 form what the groups discussed as the front end of the development process. Followed by Stage 3: The Packaging Development. Stages 4, 5, and 6 form the back end of the process, with some learnings from the development process being fed into the front end of a new FTG development project.

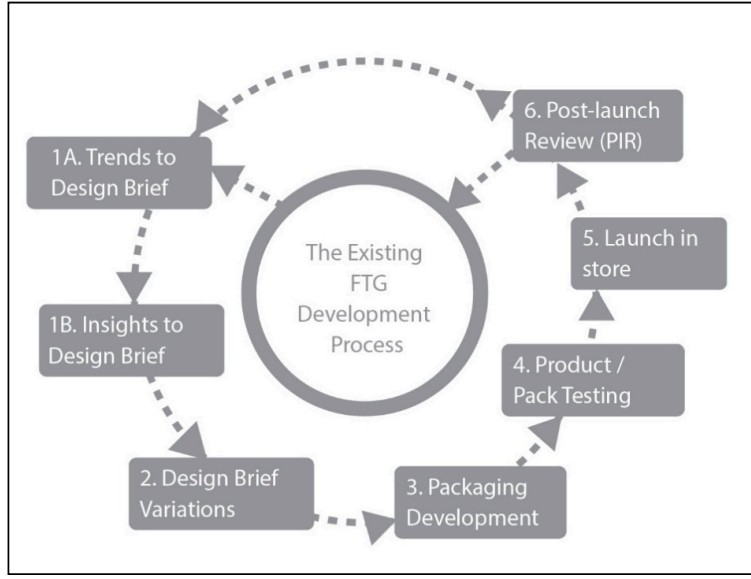

**Figure 6.** Thematic map of Task 1A analysis of existing food-to-go (FTG) development process.

In Task 1B each stakeholder group used post-it notes to plot on their business origami map where they currently engage with consumers in the development process. Table 3 summaries the existing consumer engagement with the supply chain during the FTG development process.

**Table 3.** Existing supply chain engagement with consumers during the FTG development process.

| Process Stage | Consumer Engagement | Stakeholder Quotation |
|---|---|---|
| 1B: Insights formed into a design brief | Multiple stakeholders provide sources of insights within the process to form a brief. Consumer influence on trends and insights to form new brief, often provided to the development team via the Marketing department. | *"The consumer can influence both though, because you can get insights, trends, which can then kick off an idea for both."* (FPR06) |
| 4: Product and Packaging Testing Phase | Product and Packaging factory trials occur once packaging design and costings are agreed between the Packaging Manufacturer and the Food Processor. Internal validation with multiple teams: Factory, Marketing and Kitchen trials. External supply chain trial, testing product and pack performance through the distribution chain to store. | *"But actually, can we get it through the distribution chain, does it work? Can it get into the store? Can the consumer pick it up? Will it work when they chuck it in their handbag and take it back into the office?"* (FPR01) |
| 5: Product Launch Phase | Final packaged FTG product is launched into the retail store. In rare cases this will be a soft launch in store, forming a consumer trial. In most cases the product is launched without consumer testing. | *"There's very rare cases, where if I'm looking at doing something that is outside of this process, I might initiate a trial at a specific store. I might do a trial before I roll it out."* (RET01) |
| 6: Post Launch Review | An internal review at the end of the process to evaluate the project including the product and packaging. If launch does not provide anticipated sales volumes Food Processor may conduct instore evaluation with consumers. Feedback from consumers used as insights to start a new project. | *"Particularly if we launch something that we all think is amazing and then the consumer isn't picking it up. We will go in and try and work out why they are not."* (PMN01) |

### 3.1.1. Consumer Engagement at the Front End of the Development Process

During the task, both groups discussed the interaction they have with consumers during Stage 1B in the development process, when insights are formed into a design brief. The front end of projects, also known as the insight stage, is growing in importance, with engagement from retailers, food processors, and packaging manufacturers. A packaging manufacturer explained that, *"Quite often though, early stage, before we've seen the brief and the insights, we'll do consumer panel surveys and thing like that to get their take." (PMN03)* The insights drive the project and feed into the packaging design brief as illustrated from both groups business origami maps (see Figures 4 and 5). There was also evidence of variations in how insights are used, with a retailer explaining, *"it depends it you are taking top line strategic insight, or if you are using insight to feed into what you are going to develop." (RET01)* and a packaging manufacturer explained how, *"We would also do that on new materials and things potentially, investing in consumer research probably. It would be here for us (pointing at left hand side of the business origami map), at the beginning of the process for a new range." (PMN05)*

### 3.1.2. Consumer Engagement at the Back End of the Development Process

Participants also discussed the interaction they have with consumers during the back end of the packaging development process (Stages 4 to 6), from testing the packaging prototypes to the post launch review. Stage 4 included the product and packaging testing phase, which was explained by both groups in detail. Retailers rely on food processors for packaging information and testing, with much of the validation happening internally between multiple departments. This stage checks the functional integrity of the packaging in performing its role in protecting the FTG product through the supply chain. A food processor explained how *"you've got a piece of packaging you check how it is going to run down the machines or equipment. (It is the) shelf life bit and modifications that are required." (FPR06)*

During Stage 5, the finalised packaged FTG product is launched into store. Occasionally, this will be a soft launch within a retail store, forming a trial phase with consumers, but most often there is no engagement with the consumer during this phase. A food processor described that *"If it's nothing new and we trust that it works, existing material, got the shelf life, all of that, they (consumers) wouldn't see it until it was there for them to buy." (FPR01)*

Stage 6 was defined as when the internal post-launch review occurs, evaluating stakeholder's performance during the process as well as the final product and packaging outcomes. If a product is launched and it does not sell the anticipated volumes, an instore consumer review will be conducted. This review and the learnings made from it can feed into the front end of the next packaging development project, with a food processor explaining *"we will go in and talk to people who we see picking up the product and buying it and get their feedback. Which would then kind of loop back into your insights and start that off again." (FPR01)*

### 3.1.3. Consumer Engagement Identified as Missing within the Development Process

It became clear to most stakeholders during completion of this task, especially when reviewing their business origami maps, that consumer engagement is missing within the middle of the development process; Stage 3, Packaging Development. This is the stage when design ideation, iterative design development, trialling, and sign-off of the packaging design is completed. Currently there is no engagement with the consumer within this complex looping system of design development. A food processor explained that:

> *"I think what is interesting for me coming from the task already is there is some insight that we've got from either something that the end consumer is doing or something that is happening in the world, an environmental issue or something. We get on and do all this mess. And then we go back to the end consumer when we've decided what good is." (FPR01)*

*3.2. The Need to Better Understand How Consumers Dispose of FTG Packaging out of the Home*

All participants, in both groups, agreed there is a need to better understand how consumers dispose of FTG packaging out of the home. Going forwards, the stakeholders felt the need to engage with consumers earlier in the process and how ideally, they should be testing with consumers during the whole process with a food processor explaining, *"Because you need to obviously make sure that you are developing the right packaging in the first place for the consumer." (FPR02)* There was a consensus that consumer needs are speeding up the process and the supply chain must be more reactive with a food processor clarifying that, *"Gone are the all year round projects, it's a different world. We have to react to the consumers at the end of the day, quickly don't we. Consumer needs." (FPR06)*

Consumer Behaviour Driven Packaging Development Considerations

Consumer material awareness and perception of packaging was seen by Stakeholders as the driving force for understanding how consumers dispose of FTG packaging out of the home. Stakeholders believe consumers dispose of packaging depending on the pack's predominant material and that mixed-material substrates will not be separated at disposal, with a food processor explaining that *"I think people go majority rules as well on the packaging. If it is mainly cardboard, they'll just stick it in cardboard." (FPR03)*

Stakeholders believe that more consumer insight is required to guide future selection of packaging materials. They want the information on packaging and bins to be easy to understand. A food processor stated that, *"I think everyone would be in agreement that there needs to be consumer education on materials and what's good and what's bad." (FPR02)* Stakeholders agree they need to consider end of life of packaged products and how the consumer recognises how to dispose of it before a new pack material is launched. A packaging manufacturer explained, *"We discussed it from a manufacturer's perspective, and we said that now it would always be a lead top of mind programme about end of life before we introduce any new material." (PMN05)*

Stakeholders had a clear but generalised view about consumer FTG packaging disposal behaviour out of the home. There was a consensus that FTG consumers are lazy, wanting ease when disposing of the packaging waste as explained by a food processor: *"I think people, I think consumers are and I include myself in it, we are inherently lazy, and we want everything to go in one bin and someone sort it out. Until we sort out that mindset, I think we are all a bit stuffed." (FPR05)*

Stakeholders feel that consumers are not looking for recycling option if a general waste bin is available. If there is more than one bin to dispose of the packaging, stakeholders want to understand how consumers throw FTG packs away when out of the home. Stakeholders discussed how they have witnessed a convenience and ease of disposal nature with consumers first-hand. A packaging manufacturer explained how: *"It's more than what bin, it's how they actually throw it away. I've seen people, just for convenience, they get a sandwich skillet and they shove the plastic wrap in the sandwich skillet and they put the bottle in the sandwich skillet, because it is easy and then they put this thing in the bin." (PMN03)*

*3.3. Stakeholder Evaluation of How Consumer Behaviour Methods Could be Applied*

This section will report findings from Stage 3 of the workshop where stakeholders evaluated the insight sheets found in Appendix A and discussed how the FTG packaging supply chain needs to adapt to incorporate consumer behaviour understanding within the development process.

3.3.1. Evaluation of Consumer Behaviour Research Methods

Several advantages of consumer behaviour research methods were discussed by both groups. They agreed that the contextual environment where the consumer research occurs can influence the insights gained. The retailer explained that, *"If it is an open focus group it is a very different behaviour than actually physically doing it, that's the issue isn't it. Someone might say that they're David Attenborough and then the next minute they don't really care." (RET01)*

The remote-ethnography mixed-methods approach used to gain the behavioural insights was very well received by stakeholders, described as an 'accessible' and 'insightful' tool. However, the stakeholders discussed several limitations to the research method, with some concerned about how they would fit consumer behaviour testing into the current process, due to the tight project time scales. Some stakeholders were worried about the consumer being conscious of being observed, therefore adapting their natural behaviour, leading to false insights as explained by the Recycler, *"It comes back to that key question of asking the question is totally different to what they are actually doing in real terms because of the other things that are going on. So, you physically need to get someone to do it in an environment and watch it." (REC01)* A Food Processor stated, *"It is difficult because you seem to be the experts within the industry, within your own fields. Then the snapshot of ten or twenty consumers, how real is that going to be? It's a bit of a balance really." (FPR06)* Ultimately stakeholders were keen to balance behavioural consumer insight research methods with their expert industrial packaging knowledge.

### 3.3.2. Changes Required in the Supply Chain Process

The requirement for process changes within the FTG supply chain as discussed by the stakeholders in this study, includes how to: Implement consumer testing in the process, change the way the supply chain design and test packs with consumers, and future process change requirements. The findings are illustrated in Table 4 and explained in the following subsection.

**Table 4.** Changes required to include consumers within the FTG packaging development process.

| Incorporate Behavioural Research Methods within the FTG Development Process | Alter the Supply Chain Approach to the Design and Testing of Packs with Consumers |
|---|---|
| Observe consumers within their FTG out of the home environment to gain insights. What people say versus what they do in context. | Stakeholders agree they need to conduct live pack trials with the consumer when implementing large scale change. |
| Understand how consumers dispose of FTG packaging out of the home and therefore which waste stream FTG packs currently go within. | The supply chain needs to test packs with consumer in volume, in the real context. |
| Need to improve consumer questioning to include disposal related questions. | Establishing consumer behaviour knowledge, testing existing product to guide next project within the development pipeline. |
| Place consumers in the driving seat to explain the packaging solutions they would like implemented. | Use consumer knowledge to improve the disposal process, making it a positive and easy task to complete. |
| Add further consumer workshops into process to explore product and packaging use. | Stakeholders need to work behind the scenes on the best materials to use for FTG packaging, promoting this to the consumer using clear messaging. |

### 3.3.3. Incorporating the Consumer within Future CE Packaging Development Process

During Task 3B stakeholders applied post-it notes and annotation to indicate how behavioural insights could be applied within future development processes, as illustrated in Figures 7 and 8.

Correct communication and consumer understanding of why the changes have been implemented was viewed as a vital link in the transition to future CE approaches within the supply chain. Coffee cup PR was evidenced as an example of the positive impact system change can have, as explained by a Food Processor, *"if people could see if it had a benefit (in relation to reusable coffee cups and drinks bottles). Whereas with FTG and a bin on the high street I don't think people understand that there's a benefit if they change their behaviour." (FPR01).* A packaging manufacturer suggested that there needs to be a three-way communication approach to sustainability factors describing how:

> *"You've got the consumer, but you've got the recycler and the printer with the food processor. Because I think it needs to be three-way, it needs to have all that scoping out of materials. Which is the best material sustainability wise? Can it be recycled? What are the credentials about the material recyclability?" (PMN01)*

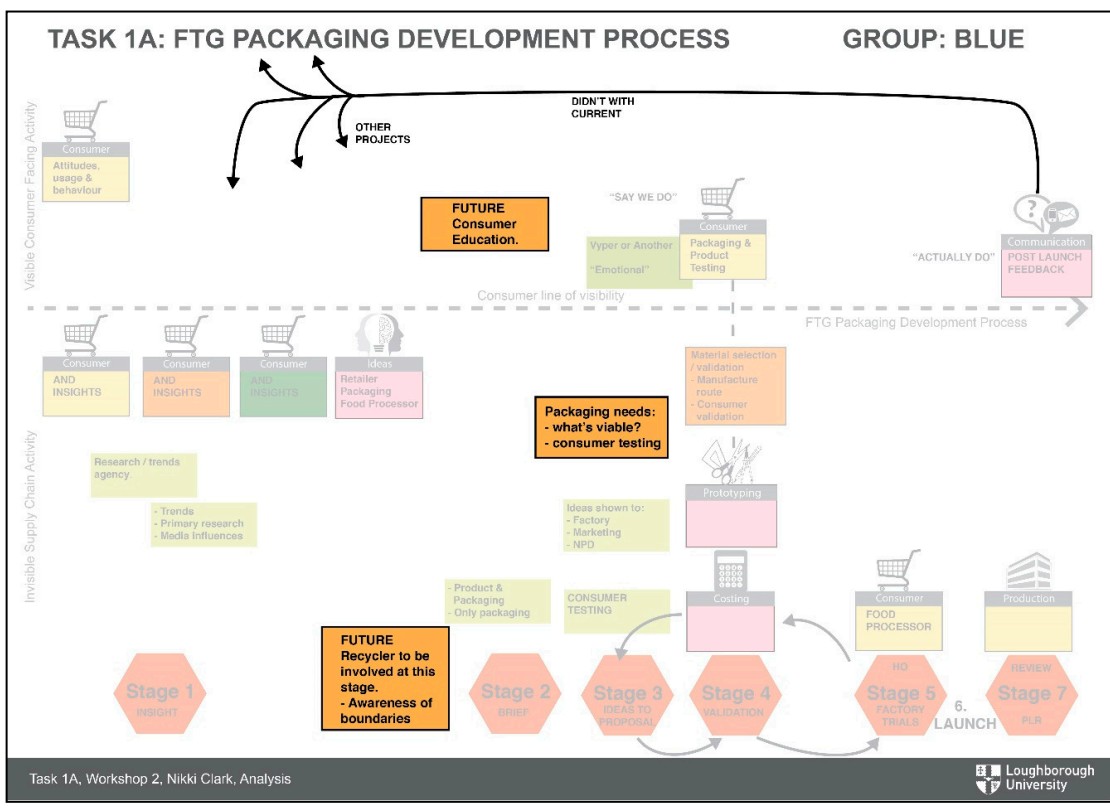

**Figure 7.** How the Blue Group would apply consumer behaviour research into the FTG packaging development process (Task 3B orange post-it notes and annotation).

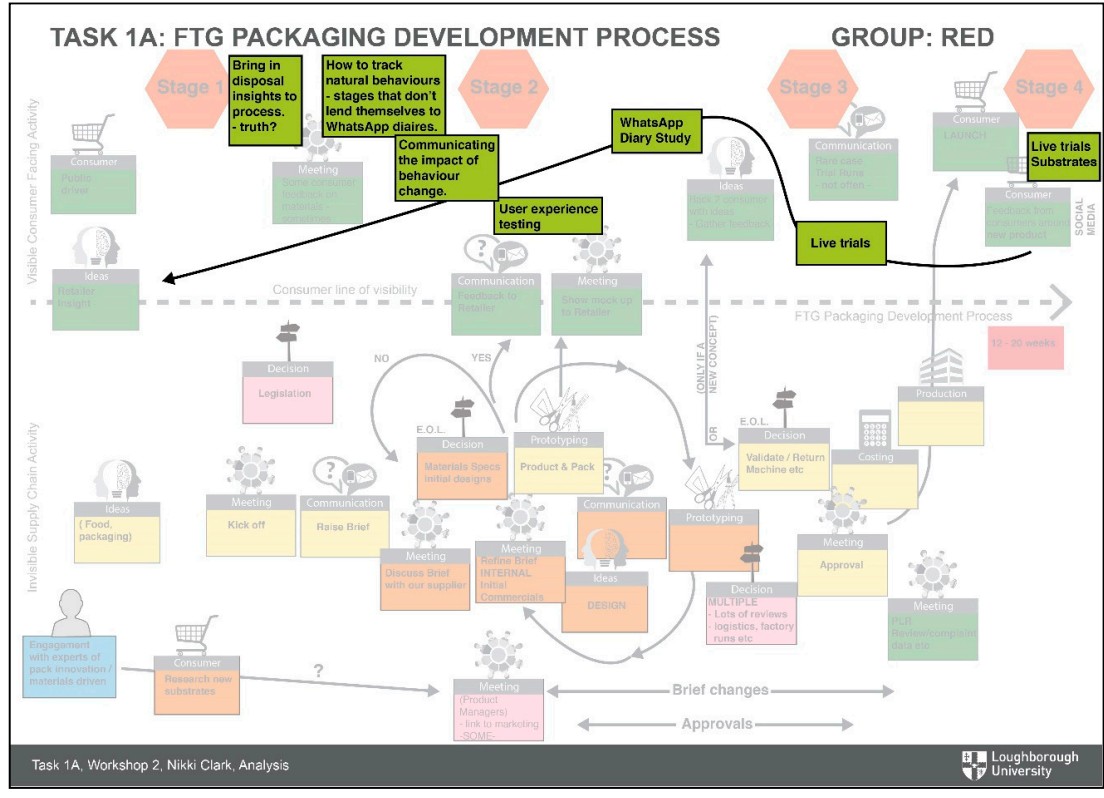

**Figure 8.** How the Red Group would apply consumer behaviour research into the FTG packaging development process (Task 3B green post-it notes and annotation).

Stakeholders discussed the use of on-pack smart technologies such as QR codes or Near Field and how these could interact with bins to aid the consumer during the disposal process. Technology was also suggested as a way of improving consumer communication whilst the disposal process is completed, with the retailer explaining how, *"It would be good if your bin only opened because your QR code for certain bins. You basically wave your pack over the bin and it wouldn't open, 'Oh I can't put it in there,' next bin 'Oh it opens, I'll put it in there.'" (RET01)* Technology was also discussed in relation to tracking packs through the existing system from retail store to user to reprocessor, with the recycler explaining that:

> *"It needs to be a wider piece of work that could be referenced to every product going through. So, we know how many are on the market. This is the pack, can you do an assessment to see physically how many of these came through the facility when you look at the funnel of where they would go to in the process? To work out the real recovery rate of that individual pack. We haven't done any of it yet, but the question has been asked a couple of times." (REC01)*

*3.4. The Strengths and Weaknesses of Incorporating Behavioural Insights into the Process*

During the final task of the workshop, stakeholders plotted the strengths and weaknesses of incorporating behavioural insights into the FTG development process onto their Group task sheet (see Figure 9). There was an even distribution of responses in both groups.

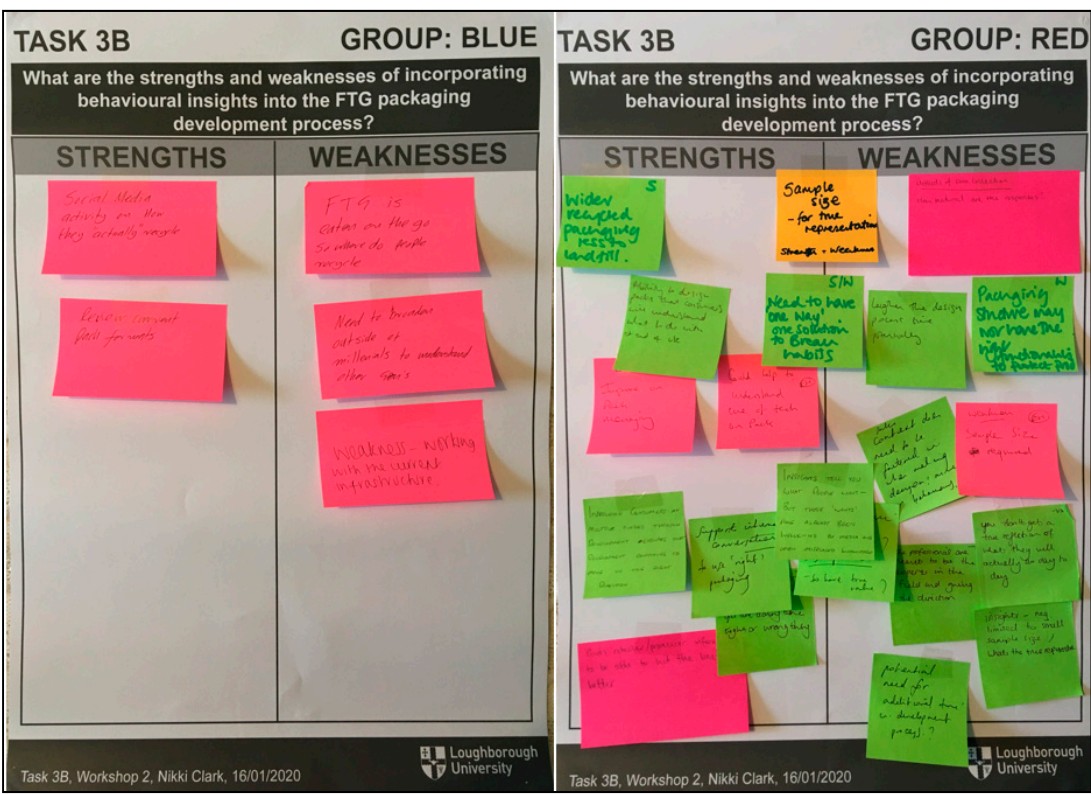

**Figure 9.** Both groups completed evaluation sheets from Task 3B.

3.4.1. Strengths of Incorporating Behavioural Insights

Most stakeholders felt remote-ethnography used as a mixed-methods research approach offered a good opportunity to gain consumer insight. By capturing unconscious consumer behaviour and then questioning them on the reasons why, as explained by a food processor, *"I thought keeping the diary and then reflecting on the diary back was really powerful." (FPR05)* However, discussions centred on how consumers know what they want, when this is often gained from media influencers, as raised by

another food processor, *"Insights are useful because you know what people want, but they only know what they want based on what they've been told before that." (FPR04)*

Stakeholders suggested applying insights to review pack formats with consumers and use this information to inform key packaging decision makers in the supply chain. The consumer disposal behaviour analysis of existing packs could be used to provide vital knowledge, influencing the front end of the design process, as explained by a food processor, *"Understand how they dispose of it this way, then put them down an actual disposal route, two different routes and how would they do it. How would we design it to be done and see where they both end up in the recycler and then pick out the key bits that need to be changed in the design?" (FPR02)*

### 3.4.2. Weaknesses of Incorporating Behavioural Insights

Stakeholders discussed the need to broaden the consumer behaviour research so that the sample size and demographic represents the whole population who purchase FTG products. Due to the scale required to achieve this, the methods of data collection may need to be altered.

*"Depends on the sample size as well. Because if you get a smaller sample size and it's kind of scattered, it's not really helpful. Or if you get a smaller group size and it's all positive, is it all just specific for that demographic?" (FPR07)*

Existing infrastructure challenges were expressed as a major weakness of using behavioural insights within the design of packaging. Stakeholders explained how they must design for the existing UK waste infrastructure, which has limited recycling bin options for FTG packaging disposal out of the home. The recycler led a group discussion on whether the supply chain just needs to accept this and design packs to fit current infrastructure, explaining that *"You can do the best you currently can. So you can do it absolutely perfectly, but if it's not in a collection policy of that area in the UK, then there is absolutely nothing you can do about it." (REC01)*

## 4. Discussion

The thematic analysis process and clustering of the data developed the following three themes; contact with consumers in the existing development process, change required in the supply chain process, and future CE approaches for the supply chain. An overview of the themes and their interrelation is illustrated in Figure 10.

All stakeholders in the co-design workshop agreed that understanding consumer behaviour is important when developing FTG products, as it is the consumer driving change within the supply chain. However, as evidenced within their business origami maps, engagement with consumers in the development process is currently limited. Only certain stakeholder groups, mainly retailers and food processors, and departments of these groups, predominantly marketing, have access to consumer insights. Currently, insights are applied at the front end of the packaging development process, aligning with the work of Nordin and Selke (2010) [33]. Industry has clear perceptions surrounding consumers and their convenience lifestyles, believing they seek ease when disposing of packaging out of the home, due to their time-poor working lives. Yet, industry stakeholders believe consumers have incorrect perceptions about packaging derived from societal influencers such as the UK media. It could be said this is a significant and damaging disconnect between industry and consumers.

Current user testing focusses on functional and aesthetic validation of packaging, often conducted online. This provides an instant emotional response, but not the detailed behavioural insights signalling how the packaging is used and disposed of within an out of the home contextual environment, and maybe more importantly lacks consumer's reasoning behind their behaviour. Investment both in process time and available capital form the current constraints in consumer testing, preventing the inclusion of behavioural research studies, run at sufficient scale, to provide meaningful and useful insights. This investment was only seen as justifiable if the project posed a business risk due to revolutionary changes to packaging formats. It would not be considered within the normal

evolutionary packaging development process. Stakeholders were concerned about how to capture consumers natural behaviour out of the home. They felt that consumer observations are vital as packaging waste and the environment has become a very emotive subject, accelerated by the influence of media, feeding incorrect information to consumers, shaping their knowledge and attitudes towards plastic packaging materials. Stakeholders are all too aware that consumers will say what researchers want to hear and that there are differences between what people say and what they do.

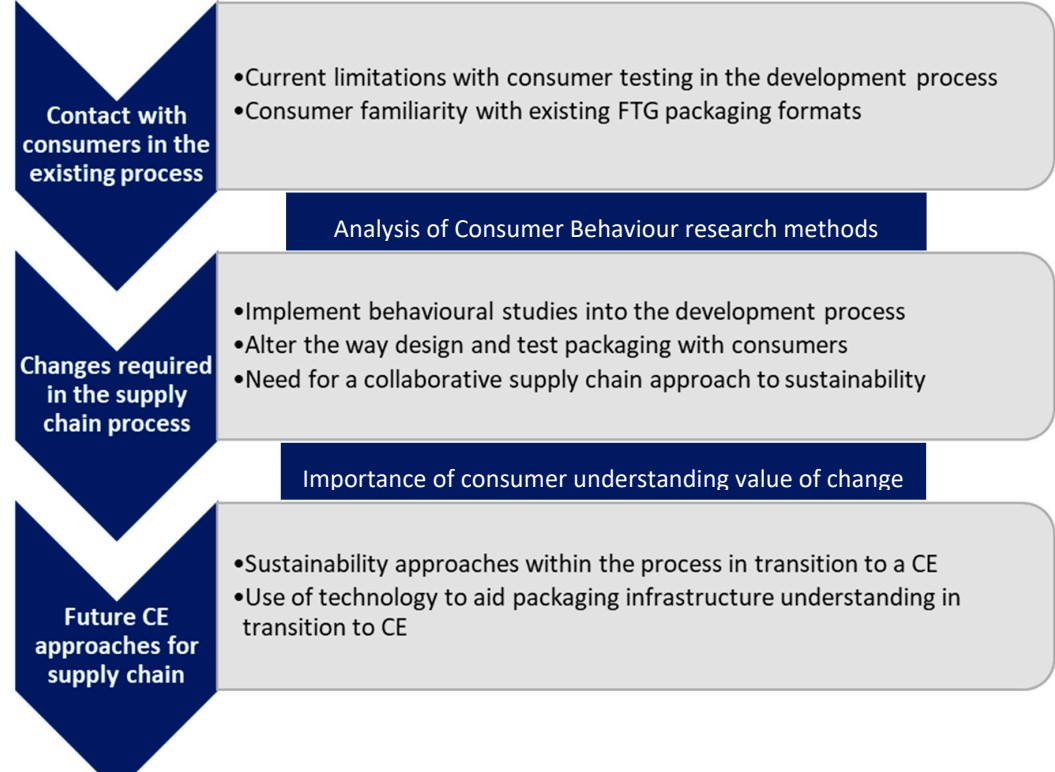

**Figure 10.** Understanding the changes in the FTG packaging development process to improve consumer behaviour disposal in a transition to a circular economy (CE).

The triangulation of mixed methods used to gain the data that formed the consumer insight sheets used in this study was viewed positively by stakeholders, providing the ability to thoroughly understand behaviours in the real contextual environment with existing packaging formats. The retail stakeholder could especially see how the methods could be applied within closed loop environments, such as the workplace, where the contextual environment remains consistent and the variations in disposal behaviours and the underlying drivers can more easily be seen. Barriers to applying behavioural studies were mainly viewed in this study as the investment both in time and money within the existing complex development process. The commercial viability of integrating the methods into the process caused concern. Investment in these techniques would most likely increase development time and financial investment across all stakeholder groups, having a knock-on effect on the cost of the packaging within the supply chain and the price of the product to the consumer. Stakeholders have additional concern about how the inclusion of consumer insights within the development process could impact negatively on packaging technology, decreasing the product shelf life, or compromising product protection, all essential for the supply chain.

Stakeholders openly admitted a need to move away from their perceptions of consumer behaviour with packaging to actual knowledge of how they will interact with different pack formats. The FTG packaging development team need to understand how the design attributes and material selection can contribute to the responsible disposal of packaging waste. They see strengths in integrating consumer

behaviour methods into the development process, especially within the complex mid-development stage, providing better understanding of disposal behaviours with packaging prototypes before approval for launch. They also identified how it could be used during the post-launch review process, evaluating the ease of disposal, with the insights being fed into the front end of the next development project. This would form feedback loops of knowledge to iteratively improve the development of FTG packaging in a transition to a CE, aligning with the findings of Lofthouse and Prendeville (2018) [34].

A collaborative approach and understanding between the stakeholder groups is required to ensure agreement on which sustainable materials and pack formats are, with the competitive nature of the retail industry placed to one side when these discussions occur. A recurring theme throughout the task was the need for a collaborative input from consumers, industry, and government. Industry and Government need to work together, with the latter driving the legislative measures to support industry in the implementation of CE systems for FTG packaging. Consumers and industry jointly need to take responsibility for changes to how single-use plastics are designed, used, and disposed (see Figure 11). The supply chain can redesign FTG packaging solutions and influence consumer choice, using existing established retailer–consumer relationships.

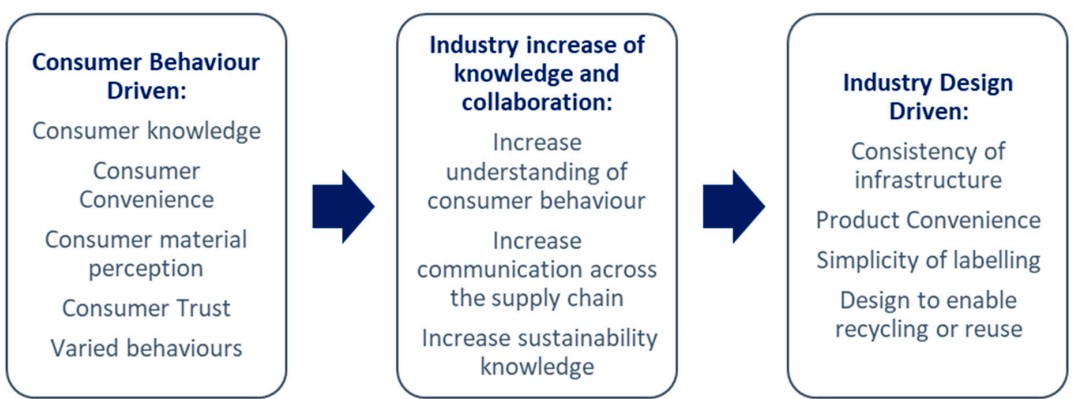

**Figure 11.** Analysis of how consumer behaviour knowledge could be applied in the FTG process.

Consumers and supply chain stakeholders could learn from each other, with industry using consumer behaviour methods to provide insights, increasing their knowledge, and understanding. This could influence both the design of sustainable packaging and how consumers improve their packaging knowledge, building trust between the two parties, enhancing their relationship. Knowledge feedback loops, as illustrated in Figure 12, between the supply chain stakeholders and consumers, appear vital in the communication and transfer of packaging change information in the transition to CE products, systems, and infrastructures. Ultimately, consumers need to understand the value and benefit of the changes implemented and industry need to provide and communicate the facts to prove the sustainable advantages.

Behavioural research methods could offer a successful and innovative approach to providing consumer insights within the supply chain. Stakeholders in this study agreed that there is a need to better understand how consumers dispose of FTG packaging out of the home and saw the strengths in using remote-ethnography mixed-methods approaches to gain insights; however, they also had some concerns about applying them (see Figure 13). Their apprehensions lay with the scale of research required to give a true representation of the target market. They raised fears about the reliance of behavioural insights to drive packaging development decisions as they may obscure other important factors such as sales context and commercial considerations of the pack and product. There may also be unintended consequences such as an increase in process time or, more worryingly, a rise in packaging or food waste. Stakeholders are also concerned that the consumer only knows what they want from packaging because of the knowledge gleaned from media influencers. Industry feel they are the experts and therefore should be instructing the consumer on what is best. This disconnect

highlights the need for increased communication and knowledge transfer between consumers and multiple stakeholders from the supply chain.

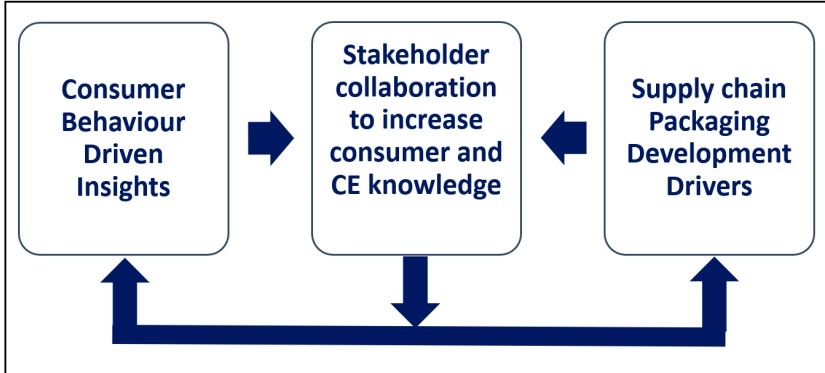

**Figure 12.** The relationship between consumer behaviour driven insights, stakeholder collaboration, and supply chain packaging development drivers.

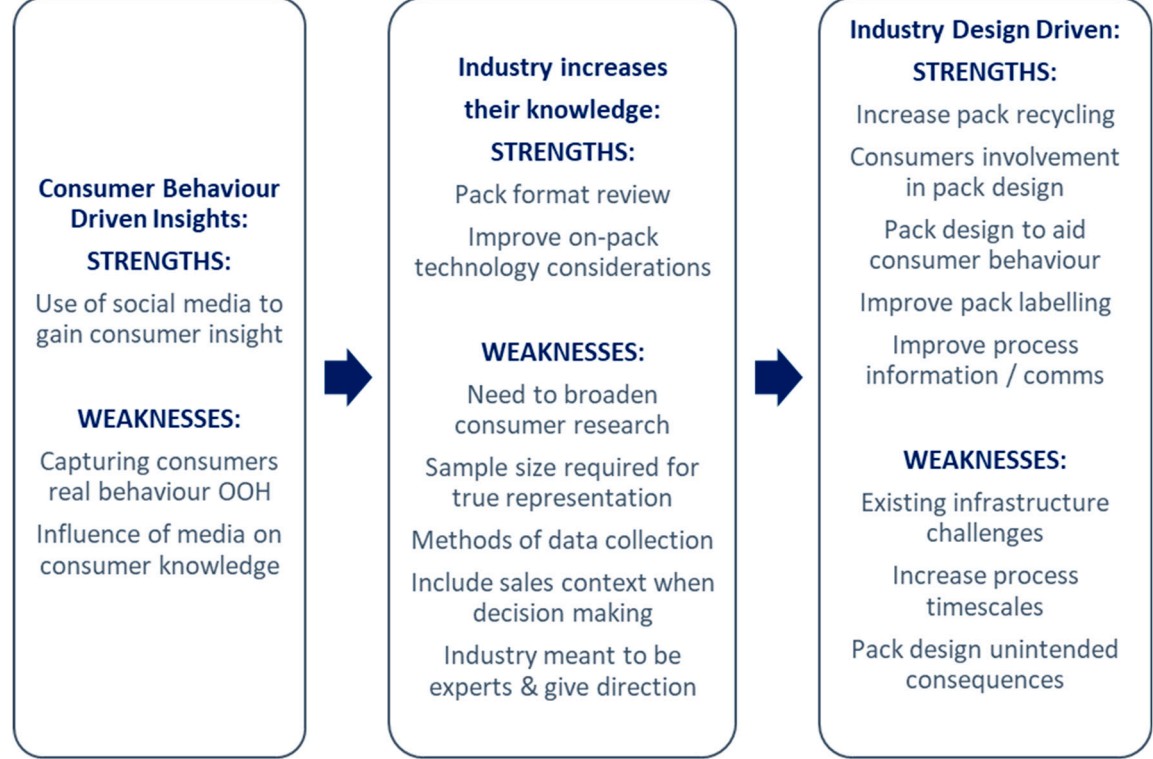

**Figure 13.** Strengths and weaknesses of incorporating behavioural insights into the FTG packaging development process.

Frustratingly for stakeholders, the greatest limitation to using behavioural studies when implementing change within the packaging industry is the restrictive UK waste infrastructure. The inconsistent processes and limited capacity of the UK waste infrastructure is a major barrier to implementing the scale of change required for circular systems to exist. The recycler representative in this study described the investment constraints facing local authorities who do not have the available budget to improve the collection reprocessing facilities necessary to circulate the volume of high-quality plastic material for circular systems to succeed. A collaborative response from Industry and the UK Government is required to implement cost-effective systems and legislative measures to ensure improvements and changes occur. Involving consumers, and their behavioural insights,

could simultaneously bring the knowledge required to change the UK's recycling infrastructure, and re-educate UK citizens about the value of packaging materials, ultimately leading to a more sustainable society. Technology was discussed as an enabler to both gain consumer insights and implement change to existing linear systems. However, robust behavioural studies providing high-quality insights could be obsolete, if the infrastructure does not exist to support the CE solutions developed using these insights.

## 5. Conclusions

Academic tools for consumer behaviour, although established for research projects, are yet to be applied to the packaging sector. The outcomes from this research showed clear benefits in firstly using mixed-methods approach to gain disposal behaviour insights, and to advance the development process using the data outcomes. Supply chain stakeholders in this study viewed the research approach to gain consumer behaviour insights as accessible. However, their concerns lie with the scale of study required to form a true picture of societies' disposal behaviour.

The main limitations to the application of the behavioural study methods within the FTG packaging development process are the time and cost to implement within an already fast-paced, commercially competitive process. This could be overcome through repeat use of the methods, iteratively improving them, so that they become both routine and valued procedures within the FTG packaging development process. In this way, behavioural research methods, and the insights they provide, become valued within the decision-making process, without being overlooked due to other project objectives. Product safety is paramount and will always be the first consideration in the development of packaging formats for food products.

An additional benefit gained from conducting the study was the collaborative nature of the stakeholder workshop, allowing knowledge from across the supply chain to be pooled, in a format that placed the competitive nature of the food packaging industry to one-side. Many stakeholders had little understanding of others involvement in the packaging development process, leading to valuable discussions and knowledge transfer during the workshop. It is a process many of the stakeholders are keen to repeat.

Currently, it is unclear which stakeholder group within the supply chain would take ownership of implementing behavioural studies into the development process. Traditionally, this would be the retailer, who drives the project through the supply chain, however this research has shown that a more collaborative approach is required and therefore a working group of multiple stakeholder representatives may be most suitable.

The research methodology used to gain the consumer disposal insights evaluated by industry stakeholders in this study must be trialled within an actual FTG packaging development process. The outputs can then be applied to developing packaging formats, which encourage a positive change in disposal behaviour. The design outcomes need to be tested with consumers to validate whether they provide the level of change in disposal behaviour required for a transition to a CE.

**Author Contributions:** Conceptualization, N.C.; methodology, N.C., R.T., and G.T.W.; formal analysis, N.C.; investigation, N.C.; writing—original draft preparation, N.C.; writing—review and editing, R.T. and G.T.W.; supervision, R.T. and G.T.W. All authors have read and agreed to the published version of the manuscript.

**Funding:** This research received no external funding.

**Conflicts of Interest:** The authors declare no conflict of interest.

## Appendix A

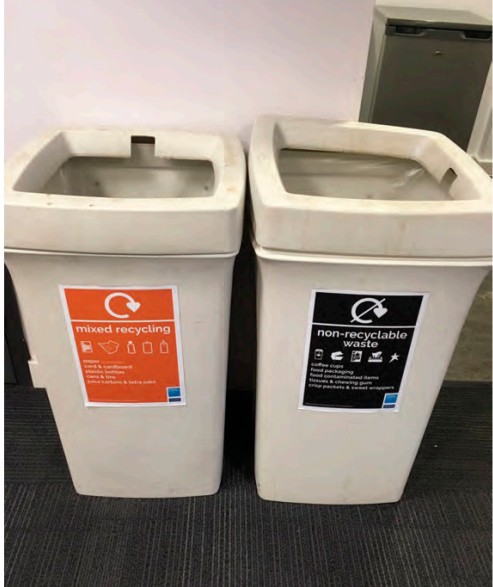

Figure A1 content:

**i1** **Origins of knowledge and current consumer disposal attitudes**

> "I do think about it quite conciously and often get confused."

> "that it is going to take 200 years to decompose, and that I can put it in a bin to be recycled or reused."

### What it means...
● Millennials have basic knowledge of the waste hierarchy, stating that reuse is preferable to recycling for plastic bottles.

● Millennials need to refresh the knowledge they gained in their childhood education twenty years ago so it is relevant for FTG packaging disposal out of the home today.

### Why it matters...
● Improve Millennials material knowledge and understanding of new technologies such as Chemical Recycling of flexibles.

● Build an appreciation of the materials value within circular systems to provide the cognitive reasoning and positive attitude to dispose of packaging correctly.

**Team Comments**

© 2020 Nikki Clark, Loughborough University

**Figure A1.** Insight sheet i1 used in Task 3A.

## i2 Impact of automaticity & habitual behaviour on disposal process

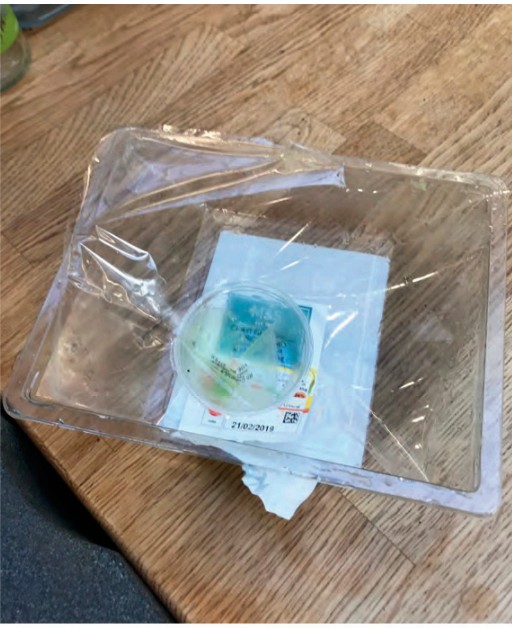

*"To conserve space in the bin. I'm not a scruncher, I'm a folder."*

*"I think everyone's default position is general waste if they are unsure."*

*"I probably thought about it afterwards, because I guess the tray could have gone in plastics, but it didn't that day."*

### What it means...

⬡ Ease of automatic behaviour conquers the knowledge of the correct disposal proffcess when time poor at work and FTG pack waste is placed into general waste when out of home.

⬡ The confusing process of recycling mixed materials requires more active thinking to influence a responsible behaviour, increasing the risk of an incorrect decision.

### Why it matters...

⬡ The influence of external contextual factors impacts on the automaticity of disposal behaviour when Millennials are time poor and unsure of what to do.

⬡ Better clarification of mixed material FTG packaging disposal is required.

### Team Comments

**Figure A2.** Insight sheet i2 used in Task 3A.

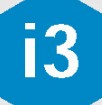

# Ease & consistency of behaviour essential for material recovery

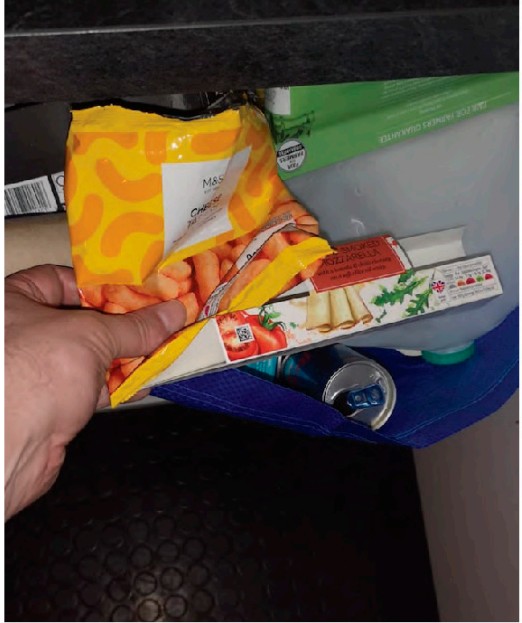

*"You should, but in day to day life no I wouldn't."*

*"I don't want to waste a lot of time and those are the bins that I have close."*

*"You know I alternate between the two. I don't know if one is right or the other."*

## What it means...

🔵 Recycling process needs to be as easy as possible so Millennials can complete the task when time poor out of the home; if easy to use it is more likely to become an automatic habitual behaviour.

🔵 Currently inconsistencies in disposal behaviour by Millennials on a daily basis, including variations in what they believe is the correct way to dispose of FTG packs.

## Why it matters...

🔵 Consider the complexity and mental effort required to dispose of a pack; easier it is the more likely it is to be done correctly.

🔵 Consistency in behaviour is required to recover the quantity and quality of materials for a Circular Economy to work effectively.

## Team Comments

© 2020 Nikki Clark, Loughborough University

**Figure A3.** Insight sheet i3 used in Task 3A.

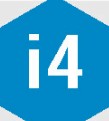

# Visible or concealed advice to others

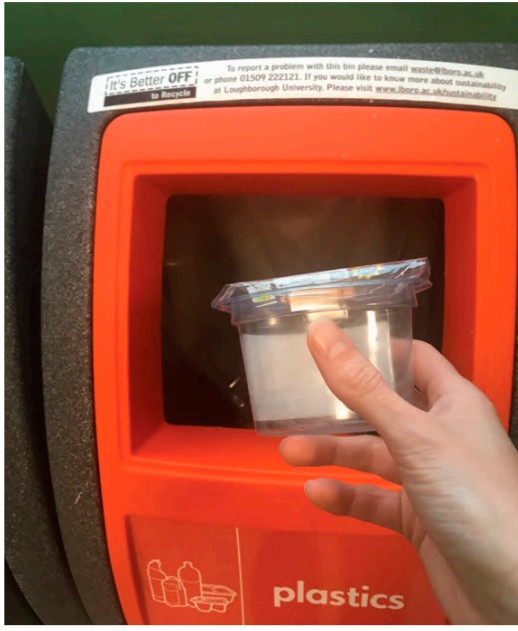

*"It really bugs me, it is contaminated, even though I'm putting mine in plastics I can see that someone has just thrown their lunch in there."*

*"If you do it everyday it just becomes a habit. I try and get everyone else to do it as well."*

## What it means...

● Millennials are keen to get others involved, sharing their knowledge with the aim to get others behaving responsible too.

● Evidence of the frustration others poor disposal behaviour has on Millennials who realise the responsible part each person in society must play in correct waste disposal.

## Why it matters...

● At home they can control the disposal behaviour of others, more challenging out of home and negates the good habitual behaviours they strive for in their lifestyle.

● Some Millennials have become waste disposal ambassadors, leading to the transfer of disposal knowledge, essential for circular systems to work effectively.

## Team Comments

© 2020 Nikki Clark, Loughborough University

**Figure A4.** Insight sheet i4 used in Task 3A.

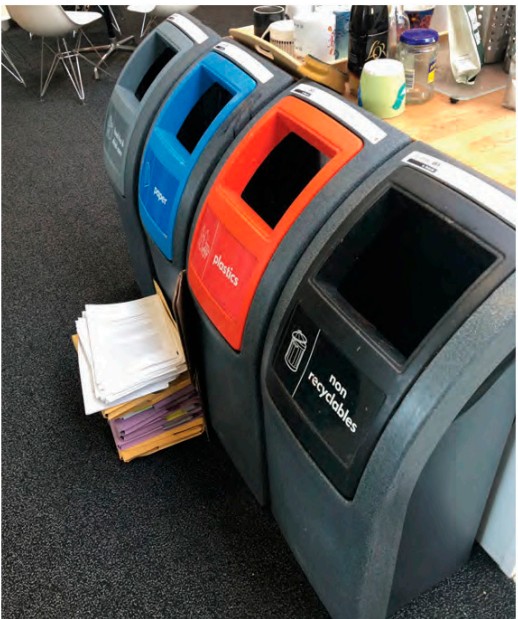

## i5 Carrot v's stick approach depends on level of self-responsibility

*"I don't think that carefully about my decisions, I just think that's a nice coloured bin."*

*"it's quite an easy task voice at the back of my head. It's quite a basic, minimal effort that you have to do to protect the environment."*

### What it means...

⬡ Millennials level of self-responsibility and their ability to reflect on their behaviour can impact on the individual's intentions at the disposal stage out of the home.

⬡ Some Millennials are unaware of their own behaviour, others realise the part their subconcious plays in their behaviour, fed by an underlying respect for the environment.

### Why it matters...

⬡ Millennials subconcious disposal knowledge, intentions and values can interrupt and rectify an incorrect action.

⬡ The level of empathy and self-acuation Millennials possess will determine whether rewarding 'carrot' or punishing 'stick' approach provides desired behaviour.

### Team Comments

**Figure A5.** Insight sheet i5 used in Task 3A.

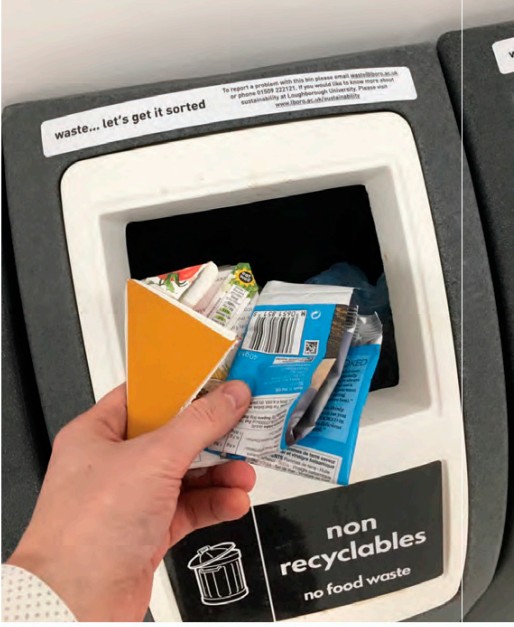

## i6  Interventions to increase level of automaticity and 'feel good' factor

*"Automatically yes, but I can't promise you it's the right decision I make."*

*"You get the person standing in front of the bin trying to decide which one their stuff needs to go into and whether it is recyclable."*

### What it means...

🔵 Millennials who stand in front of a bin confused at what is best to do shows a lack of automatic behaviour, but also the self intention to try and do the right thing.

🔵 The responsibility and 'feel good' factor to do the right thing is overruled by the convenience of not separating waste and disposing of all packaging in general waste.

### Why it matters...

🔵 There is a need for interventions to interrupt the automatic behaviour if the disposal of FTG packaging is incorrect.

🔵 Automatic habitual behaviours at the disposal stage could be harder to change using behavioural interventions, especially if they are incorrect behaviours.

### Team Comments

**Figure A6.** Insight sheet i6 used in Task 3A.

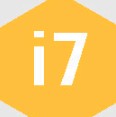

# Cleanliness of packaging impacts on recycling and water usage

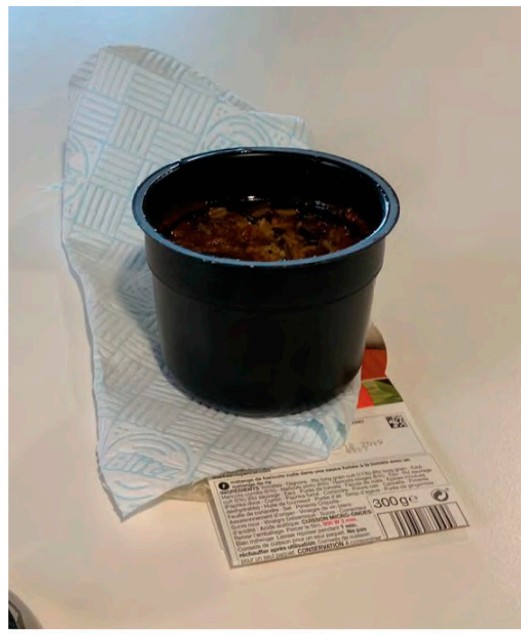

> "I would give these a rinse out first because of the yogurt in there, because when it dries it goes weird."

> "I feel there is a difference between plastic which is really dirty which would be like the black plastic and perhaps the salad plastic which is obviously really clean."

## What it means...
● Millennials disposal behaviours varied depending on how much food waste remained on FTG packaging waste.

● There were conflicting views between Millennials on whether FTG packaging needs to be washed before disposing of it within a recycling facility out of the home.

## Why it matters...
● Cleanliness of packaging at the disposal stage affects whether the pack is recycled, it could also impact refillable pack solutions introduced as part of circular systems.

● Washing packaging at the disposal stage makes the process more complex, impacts on whether the pack is recycled and wastes water, affecting sustainability.

## Team Comments

© 2020 Nikki Clark, Loughborough University

**Figure A7.** Insight sheet i7 used in Task 3A.

## i8 Pack development to increase automatic disposal behaviour

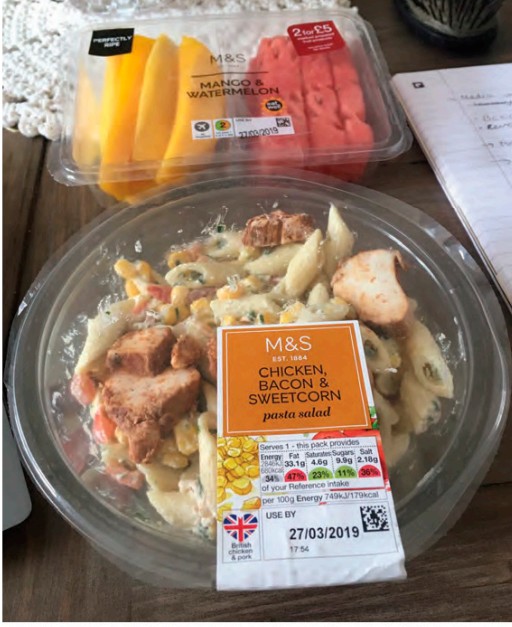

*"Because it has a plastic lining and cellophane top and you couldn't part it. So as far as I'm aware it wasn't paper if it has come in contact with food and has a plastic lining, a bit like a coffee cup."*

*"Plastics, but I'm guessing that the label should come off, but I don't know, I never actually do."*

### What it means...

🔶 Varying the materials used for FTG packaging, especially polymers, causes fluctuations in Millennials confidence of how to dispose of a pack responsibly.

🔶 Millennials have a reasonable understanding of recyclable materials, but clarity is needed for mixed-material and film-based packaging formats where separation is not an automatic behaviour.

### Why it matters...

🔶 The standardisation of packaging materials and better education of how to dispose of them could increase the level of automatic behaviour, reducing confusion and allowing the formation of correct disposal habits out of the home.

### Team Comments

**Figure A8.** Insight sheet i8 used in Task 3A.

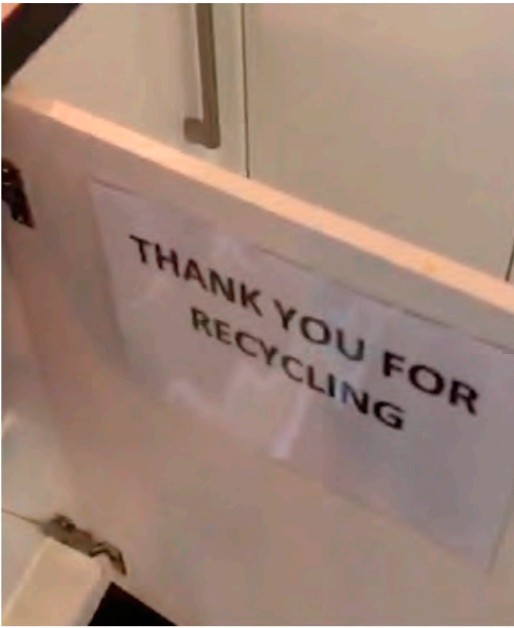

**i9 Campaigns to give pack material economic & environmental value**

*"We all have a duty to try and recycle what we can. There's a lot of wastage."*

*"You are brought up with it when you're at school, don't you, to litter pick and how to crush cans."*

## What it means...

🔶 Influence of campaigns in Millennial's childhood has instilled a sense of responsibility to protect the environment and not bury valuable materials in landfill.

🔶 Millennials currently lack knowledge of the economic importance of their disposal behaviour in a move to the circular use of packaging materials.

## Why it matters...

🔶 Industry needs to improve the economic value perception of packaging materials, especially for small pack components.

🔶 The feel-good factor of recycling needs to be nurtured and the important part Millennials play in achieving goals of a circular economy needs media publication.

**Team Comments**

© 2020 Nikki Clark, Loughborough University

**Figure A9.** Insight sheet i9 used in Task 3A.

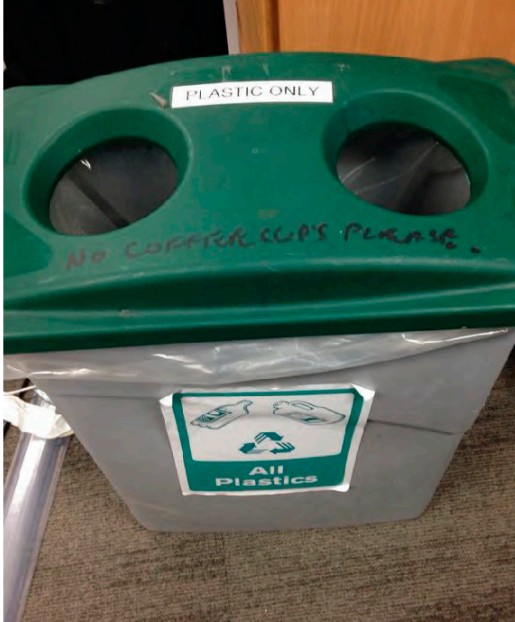

# i10 Consistency of waste systems and messaging out of the home

*"The blue says paper, but it doesn't say cardboard. Does paper allow for cardboard?"*

*"If I was to recycle this at home, I would just have thrown everything in the joint recycling. At work I would have separated it."*

## What it means...

🔶 Variations in disposal systems out of the home can lead to self-questionning of whether the correct action is being taken leading to confusion and self-doubt.

🔶 Clarity of bin communications and the consistency of waste systems is required to reduce Millennials need to adapt disposal behaviour between home and out of home.

## Why it matters...

🔶 Confusion can impact the consistency of disposal behaviours, reducing confidence and leading to mistakes & contamination.

🔶 Consistent waste processes will develop transferrable automatic behaviours, with the aim to minimise decision making, extremely important to Millennials time poor lifestyles .

## Team Comments

© 2020 Nikki Clark, Loughborough University

**Figure A10.** Insight sheet i10 used in Task 3A.

**Appendix B  (Co-design Workshop Sheets)**

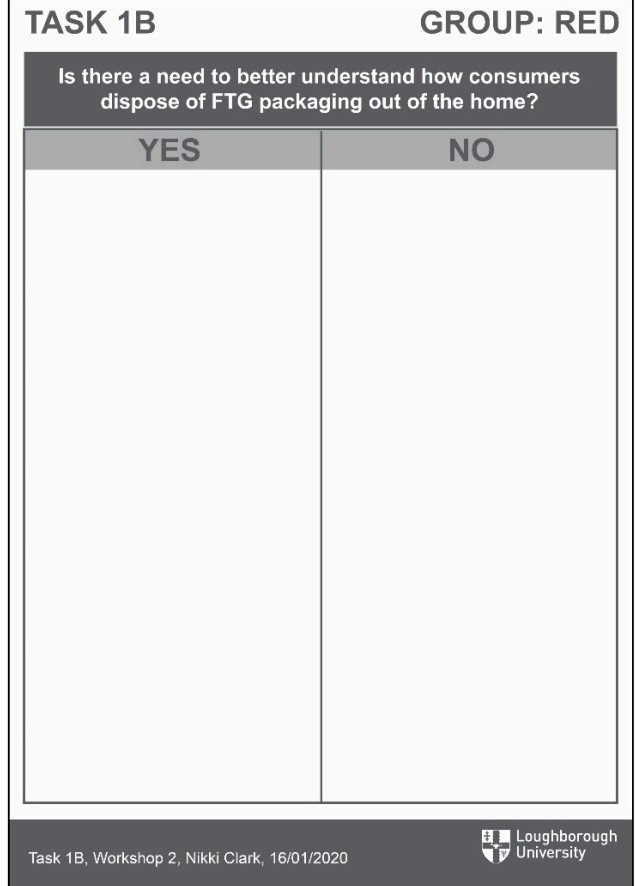

**Figure A11.** Task 1B A1 poster used in workshop; each Group provided with the same poster.

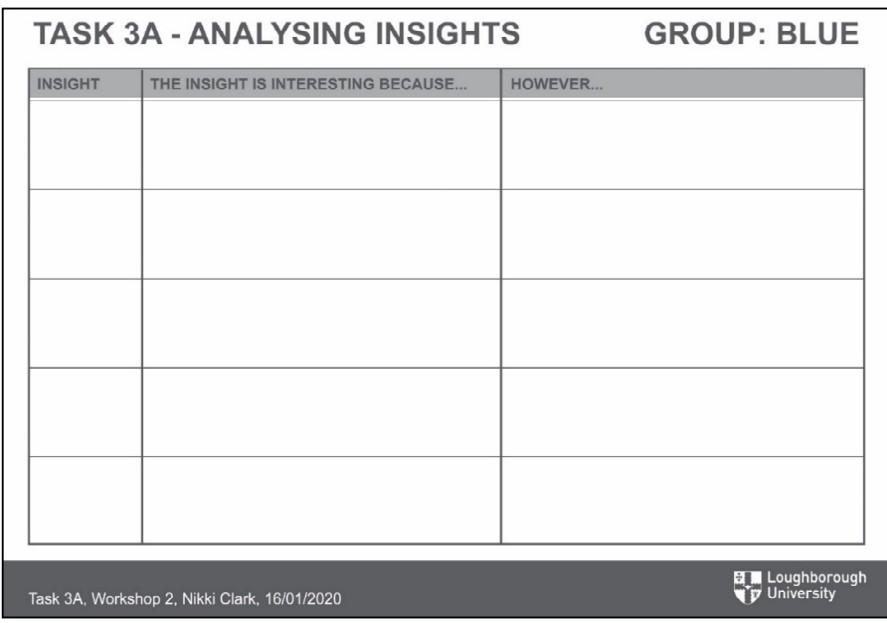

**Figure A12.** Task 3A A1 poster used in workshop; each Group provided with the same poster.

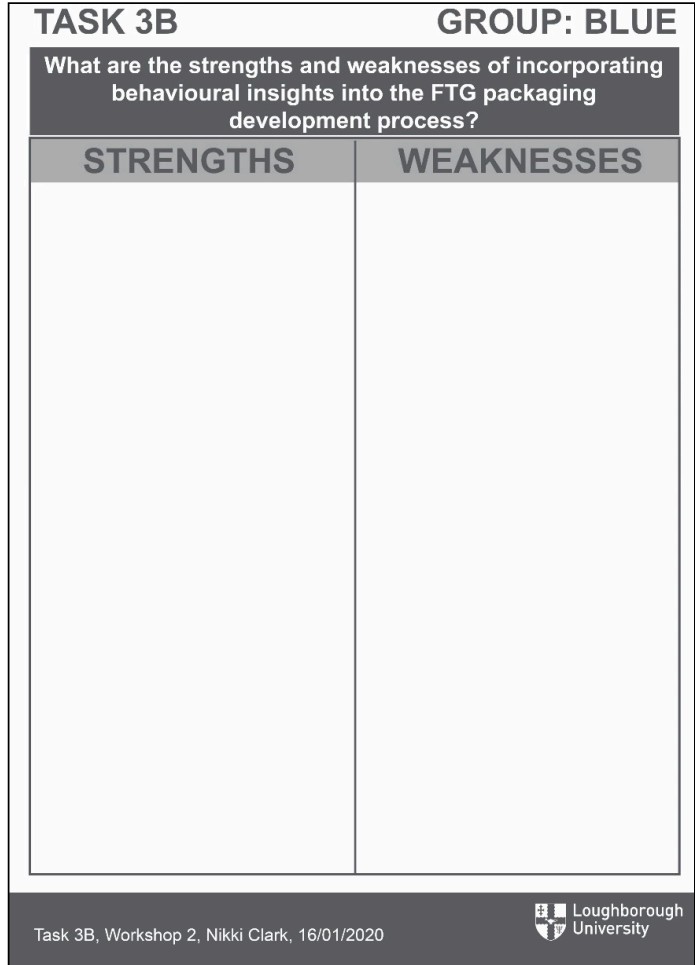

**Figure A13.** Task 3B A1 poster used in workshop; each Group provided with the same poster.

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
