# Peer review of "Incorporating Consumer Insights into the UK Food Packaging Supply Chain in the Transition to a Circular Economy"

_sustainability, doi:10.3390/su12156106_

Round 1
Reviewer 1 Report
Manuscript 878517: "Incorporating consumer insights into the UK food packaging supply chain in the transition to a circular economy".
The paper aims to evaluate the application of behaviour research methods within the UK food supply chain. The topic is very actual, interesting and aligned with the Sustainability journal aims and scope. Overall the paper is good and I think that some revisions are required to improve the research study.
Some points that should be improved:
- Within the Intoduction a detailed description of the food packaging context is provided. However some additional references and critical discussions about methods to take into account consumer behaviour seems needed.
- Novelty and scientific added-value of the paper must be better explained in the Introduction and also Abstract.
- It is not clear how the stakeholders participants have been chosen. Did the authors selected only the stakeholders who replied to the email invitation? Is the group (Table 1) statistically relevant?
- Some additional details about the methodology used to organize the co-design workshop are needed. Why the workshop was organized in three stages? Are there any references to justify this choice? Is there a scientific foundation of such organization?
- As a final part of the study (within Discussion or even Conclusions) it could be interesting to define an improved food packaging development process that takes into account consumer behaviour. This could constitute a summary of the study and a precious indication to readers that intends to use the outcomes of the present paper in real contexts.
Reviewer 2 Report
The manuscript is well written and I have no further comments to it. It is clear that the authors had put their thoughts to it. I believe that the manuscript is ready for publication as it is.
Reviewer 3 Report
The manuscript deals with an important topic of involving consumers in the design of pre-packaged Food-to-Go products in the transition to a circular economy. Research techniques used to incorporate consumer insights have been discussed thoroughly. Barriers to the application of behaviour studies have been identified. The quality of writing is high. I recommend the publication of this manuscript in the current form.
